# Towards open environments and instructions: general vision-language navigation via fast-slow interactive reasoning

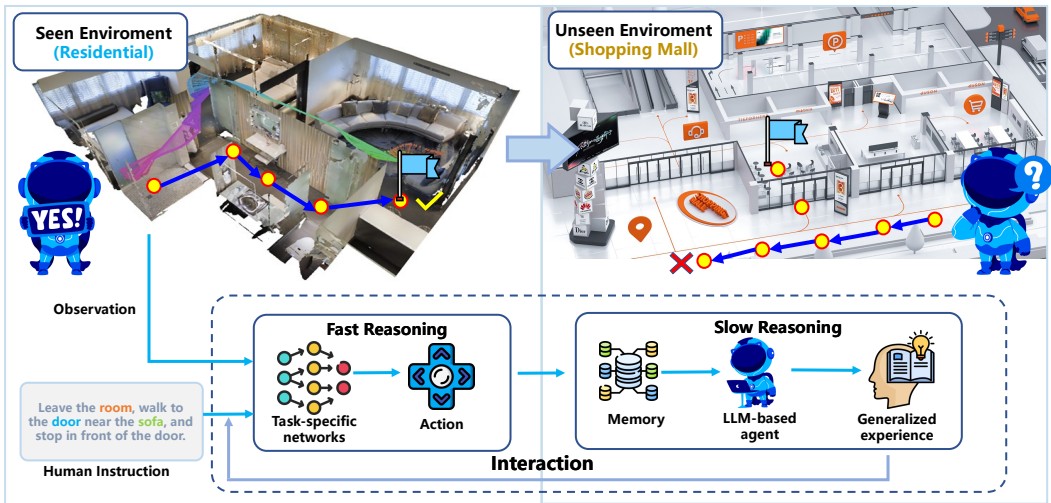

Figure 1: In the GSA-VLN task, the training set focuses on residential environments, while the test set includes non-residential scenes such as shopping malls, offices, and cinemas. It also incorporates basic, scene, and user-style instructions. The core goal is to evaluate the agent's scene generalization ability through diverse building types and instruction variations. To address this open-world navigation challenge, we propose the interactive Slow4Fast framework: "fast reasoning" is driven by a policy network that outputs actions from real-time input and stores memories; "slow reasoning" processes memories, extract generalized experiences, and reinforce the policy network.

## ABSTRACT

Vision-Language Navigation (VLN) aims to enable agents to navigate to a target location based on language instructions. Traditional VLN often follows a close-set assumption, i.e., training and test data share the same style of the input images and instructions. However, the real world is open and filled with various unseen environments, posing enormous difficulties for close-set methods. To this end, we focus on the General Scene Adaptation (GSA-VLN) task, aiming to learn generalized navigation ability by introducing diverse environments and inconsistent intructions. Towards this task, when facing unseen environments and instructions, the challenge mainly lies in how to enable the agent to dynamically produce generalized strategies during the navigation process. Recent research indicates that by means of fast and slow cognition systems, human beings could generate stable policies, which strengthen their adaptation for open world. Inspired by this idea, we propose the slow4fast-VLN, establishing a dynamic interactive fast-slow reasoning framework. The fast-reasoning module, an end-to-end strategy network, outputs actions via real-time input. It accumulates execution records in a history repository to build memory. The slow-reasoning module analyze the memories generated by the fast-reasoning module. Through deep reflection, it extracts experiences that enhance the generalization ability of decision-making. These experiences are structurally stored and used to continuously optimize the fast-reasoning

module. Unlike traditional methods that treat fast-slow reasoning as independent mechanisms, our framework enables fast-slow interaction. By leveraging the experiences from slow reasoning, it continually improves the accuracy and generalization ability of fast decisions. This interaction allows the system to continuously adapt and efficiently execute navigation tasks when facing unseen scenarios. Extensive experiments demonstrate the superiorities of our method.

# 1 INTRODUCTION

Visual-Language Navigation (VLN) (Anderson et al., 2018b) is a fundamental task in embodied AI, enabling robots to operate in real-world environments. Traditional VLN approaches like those in (Qi et al., 2020; Thomason et al., 2020), often follow a closed-set assumption, where both the training and test data share similar styles of environments and instructions. However, it fails to capture the complexity of real-world scenarios, where environments are dynamic and instructions vary widely in style and context. In practice, agents must navigate previously unseen environments, posing a significant challenge for closed-set methods, which struggle to adapt to new settings with differing environmental and instructional contexts. To this end, we focus on the recently proposed GSA-VLN (General Scene Adaptation for VLN) task (Hong et al., 2025), aiming to learn generalized navigation ability by introducing diverse environments and inconsistent intructions. Towards this task, when facing unseen environments and instructions, the challenge mainly lies in how to enable the agent to dynamically produce generalized strategies during the navigation process.

Research (Wu & Xie, 2024; Ge et al., 2023) shows that when agents transition from familiar testing environments to complex real-world settings, it can lead to spurious reasoning pathways, similar to hallucinations. This makes it difficult for agents to recognize their limitations or uncertainties (Gunjal et al., 2024; Chen et al., 2023). The root cause lies in the lack of explicit modeling of fast and slow cognitive processes, reminiscent of human *System 1* and *System 2* thinking (Yao et al., 2023; Kahneman, 2011). Recent research (Sun et al., 2025; Pan et al., 2024; Zhu et al., 2024) indicates that through fast and slow cognition systems, humans can generate stable policies that enhance their adaptation to the open world. However, existing methods often design fast and slow system as two independent, parallel systems that handle different types of tasks respectively. This fragmented structure lacks information interaction, which limits its application in the GSA-VLN task:

Although slow reasoning can solve complex scenarios, the experience it gains cannot be consolidated into the strategies of the fast reasoning network. The result is that fast reasoning always remains at its original level, and when facing similar scenarios, it still needs to repeatedly invoke slow reasoning, lacking performance improvement that evolves over time. In open worlds and unseen scenarios, generalized experience cannot be compressed into low-latency intuitive response patterns. This causes the intelligent agent to still perform like a "novice driver" in out-of-distribution scenarios, with greatly weakened generalization and adaptation capabilities.

To this end, we propose the slow4fast-VLN framework, establishing a a dynamic interactive fast-slow reasoning framework. The fast-reasoning module is an end-to-end strategy network that directly outputs actions based on real-time observations and instructions. It accumulates execution records in the history repository, building memory to face familiar environments and instructions. The slow-reasoning module refines history repository into structured knowledge by analyzing key successes and failures. Using a large language model (LLM), it reflects on and extracts generalizable experiences related to scenarios, storing them in the experience library and used to continuously optimize the fast-reasoning module. This interaction allows the system to continuously adapt and efficiently execute navigation tasks when facing unseen scenarios. Additionally, GR-DUET (Hong et al., 2025) focuses on scene adaptation from a visual perspective but overlooks adapting to diverse instruction styles. We address this by implementing instruction style transformation through Chain-of-Thought prompt engineering to capture consistent speaking styles within a fixed environment.

Our contribution can be summarized in three folds. *First*, we introduce the slow4fast-VLN framework for the GSA-VLN task, designing a dynamic interactive fast-slow reasoning framework. *Second*, we are the first to incorporate the adaptability of instruction language styles into the GSA-VLN task, filling the gap in current research regarding the diversity of language understanding. *Third*, extensive experimental results on the GSA-R2R dataset demonstrate the superiority of our method.

## 2 FAST-SLOW INTERACTIVE REASONING

### 2.1 TASK DEFINITION OF GSA-VLN

Traditional VLN requires an agent to follow a language instruction $I$ to navigate from a start viewpoint to the target viewpoint. At timestep $t$, the agent receives a panoramic observation $O_t$ containing $K$ single-view observations $O_{t,k}$, i.e., $O_t = \{O_{t,k}\}_{k=1}^{K}$. There are $N$ navigable views among $K$ views. The navigable views and a stop token $[stop]$ form the action space, from which the agent chooses one as the action prediction $a_t$. The GSA-VLN task integrates multiple datasets, covering 150 scenes and 20 building types, while clearly distinguishing between in-distribution (ID) and out-of-distribution (OOD) scenes to test the agent's ability to adapt to unfamiliar scene types. At the same time, unlike the simple instructions in regular VLN tasks, the instructions in GSA-VLN are more diverse, simulating real user language habits. They are divided into three categories: basic , scene-specific and personalized user instructions, covering various language styles.

### 2.2 OVERALL FRAMEWORK

Research (Hong et al., 2025) has shown that existing navigation methods perform poorly in OOD environments. The essential reason mainly lies in the instability of the reasoning process of the agent in complex scenarios and OOD scenarios. Therefore, the core issue is how to effectively improve the robustness of the reasoning process. In general, when facing familiar scenarios, humans rely on fast-thinking for quick decision-making, while for unfamiliar scenarios, we engage in slow-thinking to analyze and internalize the experience as a foundation for future fast thinking, which strengthens the adaptation for unknown world.

**Definition 1** *(System 1 and System 2) System 1 and System 2 are two distinct reasoning systems proposed by Daniel Kahneman in his book Thinking, Fast and Slow (Kahneman, 2011).*

*System 1 (Fast Thinking): Refers to an unconscious, automatic thinking process that is fast, intuitive, and effortless. It is responsible for automatic responses and basic cognitive operations in daily activities but is susceptible to heuristic biases and errors.*

*System 2 (Slow Thinking): Refers to a conscious, effortful thinking process that is slow, demanding, logical. It is responsible for complex calculations, reasoning, and decision-making processes.*

Inspired by this, we propose fast-slow interactive reasoning to improve the generalization of navigation (see Fig. 2). The fast-reasoning network processes real-time input, executes actions, and stores historical memory. The slow-reasoning network reflects on these memories to generate generalized experiences. These experiences guide the fast-reasoning network, providing strategic insights when faced with complex scenarios. Formally, the framework can be expressed as an iterative process:

$$\mathcal{F} = \langle \pi, R, M, A \rangle, \tag{1}$$

where $\pi$ represents the policy network for executing fast reasoning , $R$ is the reflection function, $M$ is the experience extraction and storage module, and $A$ is used to empower the fast-reasoning network with generalizable experience. The process of each episode $k$ is:

$$L_k = \pi(I_k, Env), \quad R_k = R(L_k), \quad \mathcal{E}_k = M(R_k), \quad \pi_{k+1} = A(\pi_k, \mathcal{E}_k), \tag{2}$$

where $I_k$ is the instruction of the $k$-th episode, $Env$ is the environment, $L_k$ is the generated history memory, and $\mathcal{E}_k$ is the extracted experience set.

### 2.3 FAST REASONING

Concretely, the fast-reasoning system is a policy network $\pi$, which can be any existing VLN algorithms. Here, we adopt DUET (Chen et al., 2022b) architecture. The input consists of instructions, current environmental observation (including panoramic images, GPS location, and neighbor node information), and historical navigation data. A topology mapping module dynamically constructs and updates a map with visited, navigable, and current nodes based on historical data. The global action planning module performs dual-scale encoding: the coarse-scale encoder provides global navigation scores, and the fine-scale encoder generates local actions. The dynamic fusion module then computes fusion weights to select the highest-scoring node as the next action. In addition, for

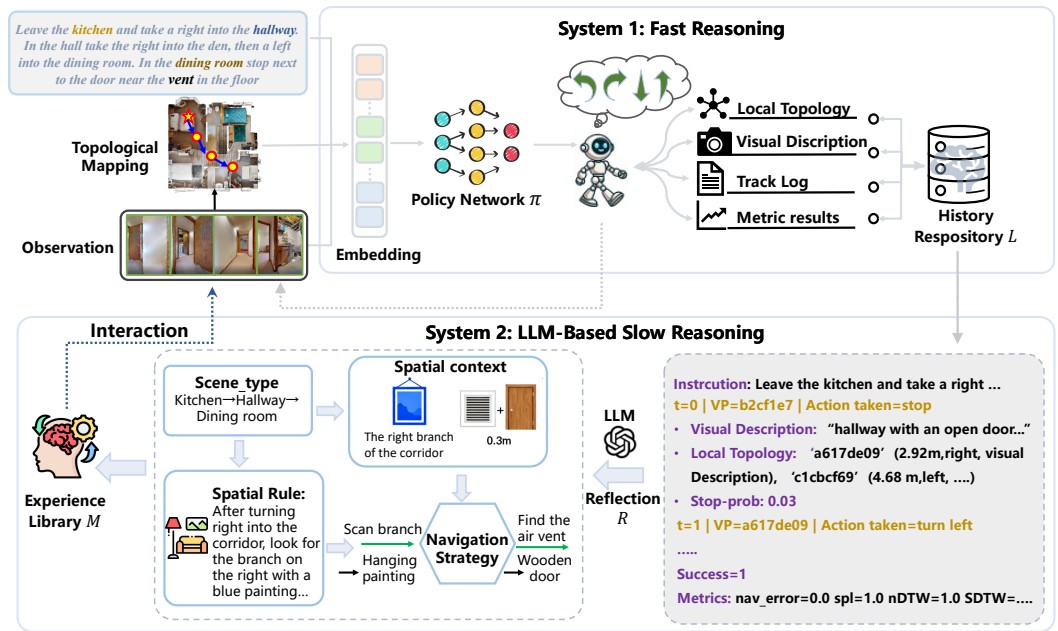

Figure 2: **Overview of our method.** The policy network processes real-time input, executes actions, and stores historical memory. The slow-reasoning network reflects on these memories to generate generalized experiences, which are then stored. These experiences guide the fast-reasoning network, providing strategic insights when faced with complex scenarios.

each node, we process its visual features using the Llama3.2-vision (Dubey & Abhinav Jauhr, 2024) to generate a textual description of the viewpoint, with each node in the topology map having its corresponding description. During navigation, a history trajectory is produced and stored as memory. A navigation episode is represented by the time step sequence $\mathcal{T} = \{t_1, t_2, \ldots, t_N\}$, where $N$ is the total number of steps. The historical data $\mathcal{L}(t_j)$ is defined as:

$$\mathcal{L}(t_j) = \left[t_j, j_{\text{seq}}, V_j, \mathcal{T}_{\text{local}}, I, A_j^s, F_v(j), \mathcal{U}_{\text{step}}\right]^\top. \tag{3}$$

For each time step $t_j$, the historical data $\mathcal{L}(t_j)$ includes the timestamp $t_j$, step sequence $j_{\text{seq}}$, viewpoint $V_j$, local topology $\mathcal{T}_{\text{local}}$ (with neighboring viewpoints, azimuth, and distance), navigation instruction $I$, selected action $A_j^s$, visual description $F_v(j)$, and step metrics $\mathcal{U}_{\text{step}}$ (such as stop probability and trajectory effectiveness). This data tracks the agent's progress throughout the episode.

While the fast reasoning module can handle navigation in most familiar scenarios, research (Wu & Xie, 2024; Ge et al., 2023) shows that when agents transition from familiar testing environments to OOD scenarios, they may generate spurious reasoning pathways. This makes it difficult for agents to recognize their own limitations or uncertainties. The fundamental reason lies in the lack of explicit modeling of slow cognitive processes. Recent research (Sun et al., 2025; Pan et al., 2024) indicates that by modeling slow cognitive systems, agents can develop stable strategies, thereby enhancing their adaptability to the open world. Therefore, we will introduce the slow-reasoning process.

## 2.4 SLOW REASONING

The slow-reasoning framework is used to convert fast-reasoning memories into structured, generalized experiences stored in the Experience Library. These experiences are integrated into the policy network for dynamic strategy adjustments, enabling continuous performance optimization.

**Experience Structure Design.** Structured experience $\mathcal{E}$ is defined as a vector:

$$\mathcal{E} = \left[S_t, C_s, R_s, T_n, \eta_s, f\right]^\top, \tag{4}$$

where $S_t$ represents the scene type, $C_s$ denotes spatial context, $R_s$ indicates spatial rules, $T_n$ is the navigation strategy, $\eta_s$ is the historical success rate, and $f$ is the frequency of occurrence. These components collectively characterize key information and knowledge in the navigation process.

**Chain-of-Thought Prompt for Reflection.** This study designs a structured chain-of-thought (CoT) reflection prompt template $\mathcal{P}$ to guide LLM in extracting valuable experience from navigation data. The template simulates human reasoning through a logical, step-by-step process:

$$\mathcal{P}(\mathcal{X}) = \mathcal{P}_{\text{intro}} + \mathcal{P}_{\text{ctx}}(\mathcal{X}) + \mathcal{P}_{\text{tasks}} + \mathcal{P}_{\text{output}}. \tag{5}$$

The template includes key components: the role definition and task guidance module $\mathcal{P}_{\text{intro}}$, which sets the LLM's role and core task; the context-filling module $\mathcal{P}_{\text{ctx}}(\mathcal{X})$, which provides necessary navigation data for analysis; the task decomposition module $\mathcal{P}_{\text{tasks}}$, which breaks down experience extraction into subtasks like scene recognition and navigation strategy analysis; and the output format constraint module $\mathcal{P}_{\text{output}}$, which ensures structured results. This prompt enhances the LLM's ability to analyze and extract valuable experience from navigation data. The detailed prompt template can be seen in Figure 5. The experience generation process is modeled as the LLM mapping function $\mathcal{F}_{\text{LLM}}$, where the input is the reflection prompt $\mathcal{P}(\mathcal{X})$ and the output is the experience $\mathcal{E}$:

$$\mathcal{E} = \mathcal{F}_{\text{LLM}}(\mathcal{P}(\mathcal{X})). \tag{6}$$

Existing methods (Sun et al., 2025; Pan et al., 2024) often design fast and slow systems as independent parallel structures for different tasks. While slow reasoning handles complex scenarios, its experience cannot integrate into fast strategies, leaving fast reasoning stagnant at the initial level. When facing similar scenarios, the slow reasoning process needs to be repeatedly invoked, which undermines the real-time navigation requirements. In fact, slow thinking should not be considered a one-time solution to complex problems. Its true value lies in produce generalized strategies that can enhance the fast-thinking system. The goal is to empower the agent to solve most problems efficiently, relying primarily on fast thinking while maintaining the flexibility to adapt to novel and unseen environments. Therefore, we establish a dynamic interactive fast-slow reasoning framework.

## 2.5 INTERACTION BETWEEN FAST-SLOW REASONING

The fast reasoning module processes real-time navigation inputs and stores historical memories. Then, the slow reasoning module reflects on and summarizes them, extracting generalizable experiences to deal with OOD scenarios. The key here is how to achieve fast-slow interaction? How to use slow reasoning to empower fast reasoning networks. To this end, we adopt the following solution: the slow reasoning module retrieves experiences related to the current scenario from the experience library and encodes them into specific vectors. Next, the visual features of the fast reasoning network $\pi$ are fused with this vector through attention. And the experience-enhanced navigation decisions are output, which is the interaction between fast and slow reasoning.

**Experience Retrieval and Encoding**. The experience library $M = \{E_1, E_2, \ldots, E_K\}$ is a finite set of experiences with a capacity of $K$. At the current timestep of navigation decision-making, let the current context be $\mathcal{X}_{\text{cur}}$ (including information such as the current scene, spatial location). First, extract the retrieval key features $\mathcal{K} = [S_t^{\text{cur}}, C_s^{\text{cur}}, T_n^{\text{cur}}]$ from $\mathcal{X}_{\text{cur}}$, then calculate the feature similarity $\text{sim}(\mathcal{K}, \mathcal{E}_i)$ between $\mathcal{K}$ and all $\mathcal{E}_i$ in the $M$ (as Equation 10). Let the retrieval threshold be $\tau_{\text{retrieve}}$, and select the experiences with a similarity greater than or equal to $\tau_{\text{retrieve}}$, sorted in descending order of similarity. The most relevant $M$ experiences are selected to form the experience set $M_{\text{sel}}$. In order to align the experiences with the feature of the fast reasoning network $\pi$, an experience encoder $G_{\text{enc}}$ is designed to convert each selected experience $E_{\text{sel},k}$ into a vector representation $F_e(k) \in \mathbb{R}^d$, where $d$ is the experience embedding dimension. For discrete features $S_t, C_s, T_n$, they are converted into vectors through an embedding layer, with embedding dimension $d/3$. A linear layer and activation function are applied to obtain the final experience embedding. For the $M$ experiences in $M_{\text{sel}}$, after encoding, the set of experience features $F_e = \{F_e(1), \ldots, F_e(M)\}$ is obtained.

**Experience Fusion**. Let the original visual feature of the fast reasoning network $\pi$ be $F_v \in \mathbb{R}^{B \times L \times D}$ (where $B$ is the batch size, $L$ is the number of views, and $D$ is the dimension of visual features). The core of the interaction between fast and slow reasoning is to fuse $F_v$ and $F_e$ through an attention mechanism. First, we expand $F_e$ to a dimension consistent with the batch size of $F_v$, resulting in $F_e^{\text{exp}} \in \mathbb{R}^{B \times M \times d}$. Then, we calculate the attention weights between visual features and experience features using a multi-head attention layer:

$$F_{\text{att}}, \omega = \text{MultiHeadAttn}(Q = F_v, K = F_e^{\text{exp}}, V = F_e^{\text{exp}}), \tag{7}$$

where $\omega$ is the attention weight, and $F_{\text{att}} \in \mathbb{R}^{B \times L \times d}$ is the experience feature after attention weighting. Next, we perform a concatenation operation on $F_v$ and $F_{\text{att}}$ along the feature dimension, and

map the result back to the feature dimension of fast reasoning network $\pi$ via a linear layer:

$$F_{\text{fused}} = \sigma \left( W_{\text{fusion}} \cdot [F_v; F_{\text{att}}] + b_{\text{fusion}} \right), \tag{8}$$

where $W_{\text{fusion}} \in \mathbb{R}^{D \times (D+d)}$ and $b_{\text{fusion}} \in \mathbb{R}^D$ are the parameters of the fusion layer. Finally, we replace the original visual feature of the fast reasoning network $\pi$ with $F_{\text{fused}}$, and input it into the model for forward computation to obtain the experience-empowered navigation decision output $Y_{\text{enhanced}}$:

$$Y_{\text{enhanced}} = \pi(F_{\text{fused}}, I), \tag{9}$$

where $I$ is the navigation instruction and $Y_{\text{enhanced}}$ including the action probability distribution and navigation confidence.

### 2.6 INSTRUCTION STYLE CONVERSION

Using a LLM model as the foundation, instruction style conversion is implemented through Chain-of-Thought prompt engineering. The specific process is as follows: when the system receives User or Scene style instructions, it first constructs a prompt text containing style conversion requirements. This prompt is then input into the LLM model, which automatically identifies and converts stylistic features in the instructions based on its language understanding capabilities, while preserving the core navigation semantics of the instruction unchanged. Finally, it outputs the converted Basic-style instruction. The system also computes a confidence score for the conversion; if the confidence exceeds a preset threshold, the converted instruction is used, otherwise the original instruction is retained. This entire process occurs in real-time during navigation training without requiring additional pre-training phases. Through this approach, dynamic conversion from Scene and User style instructions to Basic style is achieved, providing the navigation model with uniformly formatted instruction inputs. The detailed prompt template content can be seen in Figure 6 and Figure 7.

## 3 EXPERIMENTS

### 3.1 EXPERIMENTAL SETUP

**Datasets and Evaluation.** We follow the benchmark proposed by GR-DUET (Hong et al., 2025) and use the GSA-R2R dataset, which combines data from Habitat-Matterport3D (HM3D) and Matterport3D (MP3D). The dataset contains 150 evaluation scenes, including 75 in-distribution (ID) residential scenes and 75 out-of-distribution (OOD) non-residential scenes across 19 categories. In terms of language instructions, they include Basic instructions, as well as Scene instructions and User-style instructions generated by LLM simulating typical scene users or TV drama characters. For 600 paths in each scene, 7 instruction styles are generated, ultimately resulting in 90,000 path-instruction pairs. The splits are named using the format "Val/Test-R/N-Basic/Scene/User", where "R" denotes residential and "N" represents non-residential scenes. We therefore adopt several evaluation metrics for navigation, including Navigation Error (NE, the distance between agent's final location and the target location), Success Rate (SR), and SR penalized by Path Length (SPL), Trajectory Length (TL, the total navigation distance in meters), Normalized Dynamic Time Warping (nDTW, a measure of instruction fidelity by computing the similarity between the reference path and the predicted path). More details about GSA-R2R dataset can be seen in Section A.7.

**Implementation Details.** For the fast-reasoning component, we use the DUET (Chen et al., 2022b) architecture. Image features are extracted using CLIP-ViT-B/16, and we employ 9 transformer layers in the text encoder. Other hyperparameters are set the same as in GR-DUET. What differs is that the visual observations obtained are converted into textual descriptions using llama3.2-vision (Dubey & Abhinav Jauhr, 2024). The entire slow-reasoning module is based on llama3.2-vision.

### 3.2 EXPERIMENTAL RESULTS

**Environment Adaptation.** We first tested these adaptation methods using basic instructions in different environments, with the results shown in Table 1. Compared to our baseline method GR-DUET (Hong et al., 2025), our method achieved the best performance on both the residential (R) and non-residential (N) datasets, with success rates (SR) improving by 1.5 % and 2.2%, respectively. This indicates that our fast reasoning module accumulates long-term memories related to scenarios,

Table 1: Comparison of different adaptation methods in GSA-R2R with basic instructions.

| Methods | Test-R-Basic | | | | | Test-N-Basic | | | | |
|---|---|---|---|---|---|---|---|---|---|---|
| | TL | NE↓ | SR↑ | SPL↑ | nDTW↑ | TL | NE↓ | SR↑ | SPL↑ | nDTW↑ |
| *Baseline* | | | | | | | | | | |
| DUET (Chen et al., 2022b) | 13.1 | 4.2 | 57.7 | 47.0 | 55.6 | 14.8 | 5.3 | 48.1 | 37.3 | 45.9 |
| *Optimization-Based Methods* | | | | | | | | | | |
| +MLM (Devlin, 2018) | 13.1 ±0.1 | 4.1 ±0.1 | 57.9 ±0.2 | 47.3 ±0.1 | 55.9 ±0.2 | 13.1 ±0.2 | 5.3 ±0.1 | 48.3 ±0.5 | 38.8 ±0.5 | 48.4 ±0.3 |
| +MRC (Lu et al., 2019) | 13.1 ±0.1 | 4.2 ±0.1 | 57.7 ±0.1 | 47.0 ±0.1 | 55.6 ±0.1 | 14.7 ±0.1 | 5.3 ±0.1 | 48.1 ±0.1 | 37.3 ±0.1 | 45.9 ±0.1 |
| +BT (Wang et al., 2020) | 8.0 ±0.1 | 3.8 ±0.1 | 61.3 ±0.6 | 57.7 ±0.3 | 70.1 ±0.5 | 7.9 ±0.0 | 5.2 ±0.1 | 49.5 ±0.8 | 46.0 ±0.8 | 59.4 ±0.9 |
| +TENT (Wang et al., 2021) | 14.6 ±0.0 | 4.2 ±0.0 | 57.2 ±0.4 | 44.2 ±0.4 | 52.9 ±0.1 | 16.2 ±0.1 | 5.4 ±0.1 | 46.5 ±0.4 | 33.7 ±0.2 | 42.6 ±0.3 |
| +SAR (Niu et al., 2023) | 13.8 ±0.8 | 4.0 ±0.1 | 57.6 ±0.2 | 44.6 ±0.2 | 53.0 ±0.2 | 16.5 ±0.0 | 5.4 ±0.0 | 44.6 ±1.5 | 31.5 ±1.6 | 40.6 ±1.3 |
| *Memory-Based Methods* | | | | | | | | | | |
| TourHAMT (Krantz et al., 2023) | 11.6 ±0.1 | 7.4 ±0.1 | 14.9 ±0.1 | 12.2 ±0.1 | 34.7 ±0.1 | 9.4 ±0.1 | 7.7 ±0.1 | 11.0 ±0.2 | 8.6 ±0.2 | 32.2 ±0.1 |
| OVER-NAV (Zhao et al., 2024) | 14.1 ±0.1 | 6.7 ±0.0 | 22.3 ±0.3 | 16.8 ±0.2 | 37.1 ±0.1 | 11.4 ±0.1 | 7.1 ±0.1 | 16.6 ±0.2 | 13.0 ±0.1 | 35.0 ±0.2 |
| GR-DUET (Hong et al., 2025) | 9.4 ±0.0 | 3.1 ±0.0 | 69.3 ±0.2 | 64.3 ±0.1 | 71.4 ±0.1 | 8.9 ±0.0 | 4.4 ±0.0 | 56.6 ±0.1 | 51.5 ±0.1 | 61.0 ±0.1 |
| Ours | 9.6 ±0.1 | **2.9** ±0.2 | **70.8** ±0.1 | **65.0** ±0.1 | **72.1** ±0.1 | 10.2 ±0.1 | **4.2** ±0.0 | **58.4** ±0.2 | **52.9** ±0.1 | **62.4** ±0.3 |

Table 2: Comparison of different adaptation methods in GSA-R2R with User instructions.

| Methods | Child | | Keith | | Moira | | Rachel | | Sheldon | |
|---|---|---|---|---|---|---|---|---|---|---|
| | SR↑ | SPL↑ | SR↑ | SPL↑ | SR↑ | SPL↑ | SR↑ | SPL↑ | SR↑ | SPL↑ |
| *Baseline* | | | | | | | | | | |
| DUET | 54.3 | 44.1 | 56.0 | 46.3 | 52.3 | 43.3 | 56.3 | 46.4 | 54.0 | 44.4 |
| *Optimization-Based Methods* | | | | | | | | | | |
| +MLM | 54.5 ±0.2 | 44.7 ±0.2 | 56.4 ±0.3 | 46.8 ±0.3 | 53.8 ±0.3 | 43.6 ±0.4 | 56.8 ±0.5 | 46.6 ±0.6 | 54.5 ±0.4 | 44.2 ±0.3 |
| +MRC | 54.4 ±0.2 | 44.2 ±0.1 | 56.0 ±0.1 | 46.3 ±0.1 | 52.3 ±0.2 | 43.3 ±0.1 | 56.0 ±0.1 | 46.2 ±0.2 | 53.7 ±0.2 | 44.2 ±0.4 |
| +BT | 57.5 ±0.7 | 54.0 ±0.9 | 61.2 ±0.3 | 57.9 ±0.4 | 57.3 ±0.5 | 54.0 ±0.6 | 61.6 ±0.8 | 58.1 ±0.7 | 57.6 ±0.5 | 54.3 ±0.5 |
| +TENT | 54.3 ±0.2 | 41.7 ±0.1 | 55.4 ±0.2 | 43.8 ±0.2 | 51.7 ±0.2 | 41.0 ±0.1 | 55.0 ±0.2 | 43.2 ±0.2 | 53.0 ±0.2 | 41.9 ±0.1 |
| +SAR | 54.5 ±0.5 | 41.5 ±0.4 | 54.9 ±0.3 | 43.1 ±0.2 | 51.0 ±0.4 | 40.3 ±0.6 | 55.3 ±0.5 | 43.0 ±0.6 | 52.9 ±0.2 | 41.4 ±0.4 |
| *Memory-Based Methods* | | | | | | | | | | |
| TourHAMT | 14.6 ±0.2 | 12.0 ±0.2 | 15.1 ±0.2 | 12.3 ±0.1 | 13.9 ±0.1 | 11.3 ±0.1 | 15.3 ±0.1 | 12.5 ±0.1 | 14.4 ±0.1 | 11.8 ±0.1 |
| OVER-NAV | 20.9 ±0.4 | 16.1 ±0.2 | 20.5 ±0.1 | 16.4 ±0.1 | 19.5 ±0.2 | 15.4 ±0.2 | 20.6 ±0.3 | 16.2 ±0.2 | 20.5 ±0.1 | 16.2 ±0.1 |
| GR-DUET | 65.2 ±0.1 | 59.7 ±0.1 | 66.7 ±0.1 | 62.0 ±0.1 | 60.9 ±0.2 | **56.2** ±0.2 | 67.1 ±0.1 | 62.2 ±0.1 | 63.9 ±0.1 | 58.9 ±0.1 |
| Ours | **65.5** ±0.3 | **60.4** ±0.1 | **68.3** ±0.2 | **62.3** ±0.1 | **62.3** ±0.2 | 55.4 ±0.2 | **68.6** ±0.1 | **62.3** ±0.1 | **65.5** ±0.1 | **61.1** ±0.1 |

and the slow reasoning module refines them into generalized scenario navigation rules and strategies, thereby empowering fast reasoning, helping the agent adapt to both in-distribution (ID) and out-of-distribution (OOD) environments.

**Instruction Adaptation.** We evaluated these methods under different instruction styles. Table 2 presents the results for the model under five user instructions roles, while Table 3 shows their performance with scene instructions. First, DUET's (Chen et al., 2022b) performance data indicates that different expression styles introduce varying levels of difficulty in instruction interpretation for VLN models. Second, our method outperforms GR-DUET (Hong et al., 2025) in both scene and user-style instructions. The key innovation lies in addressing GR-DUET's limitation in adapting to instruction styles. We achieve this through an LLM-based instruction style conversion mechanism (us-

Table 3: Comparison of different adaptation methods in GSA-R2R with Scene instructions.

| Methods | Test-N-Scene | | | | |
|---|---|---|---|---|---|
| | TL | NE↓ | SR↑ | SPL↑ | nDTW↑ |
| *Baseline* | | | | | |
| DUET | 14.9 | 6.4 | 39.6 | 30.1 | 40.9 |
| *Optimization-Based Methods* | | | | | |
| +MLM | 14.3 ±0.1 | 6.5 ±0.1 | 39.8 ±0.1 | 30.5 ±0.1 | 41.1 ±0.1 |
| +MRC | 14.9 ±0.1 | 6.4 ±0.1 | 39.7 ±0.1 | 30.2 ±0.1 | 40.9 ±0.1 |
| +BT | 8.4 ±0.0 | 6.3 ±0.2 | 41.2 ±1.5 | 38.2 ±1.2 | 51.3 ±1.2 |
| +TENT | 16.4 ±0.1 | 6.3 ±0.1 | 40.6 ±0.2 | 28.9 ±0.2 | 38.9 ±0.2 |
| +SAR | 16.3 ±0.1 | 6.0 ±0.2 | 41.4 ±0.6 | 29.1 ±0.3 | 39.0 ±0.3 |
| *Memory-Based Methods* | | | | | |
| TourHAMT | 7.3 ±0.1 | 8.1 ±0.1 | 9.7 ±0.1 | 8.0 ±0.1 | 32.3 ±0.1 |
| OVER-NAV | 11.8 ±0.1 | 7.6 ±0.2 | 16.7 ±0.4 | 12.6 ±0.2 | 34.6 ±0.3 |
| GR-DUET | 10.1 ±0.0 | 5.5 ±0.0 | 48.1 ±0.1 | 42.8 ±0.1 | 53.7 ±0.1 |
| Ours | 8.9 ±0.2 | **5.1** ±0.0 | **50.7** ±0.1 | **46.6** ±0.1 | **57.8** ±0.3 |

ing prompt engineering to transform scene/user styles into the model's familiar Basic style while retaining core semantics). Meanwhile, we incorporated a dynamic feedback loop where quick reasoning accumulates "scene-instruction-action" memories, while slow reasoning refines structured knowledge, which then feeds back into quick decision-making. As a result, our approach achieves better navigation success rates, path accuracy, and other metrics under both instruction styles.

### 3.2.1 ABLATION STUDY

**Component Analysis.** The main modules we propose consist of two parts: the Fast-Slow Reasoning (FSR) framework and Instruction Style Conversion (ISC). We conducted ablation experiments on them, as shown in Table 4. First, compared with the first row, when ISC is added in the second row, it can be seen that Test-R-Basic and Test-N-Basic remain unchanged, while the performance of

Table 4: Analysis of ablation experiments on each module.

| FSR | ISC | Test-R-Basic | | Test-N-Basic | | Test-N-Scene | |
|---|---|---|---|---|---|---|---|
| | | SR↑ | SPL↑ | SR↑ | SPL↑ | SR↑ | SPL↑ |
| × | × | 64.0 ±0.1 | 58.0 ±0.2 | 53.7 ±0.2 | 47.5 ±0.1 | 42.4 ±0.1 | 42.8 ±0.2 |
| × | ✓ | 64.0 ±0.1 | 58.0 ±0.2 | 53.7 ±0.2 | 47.5 ±0.1 | 46.1 ±0.4 | 44.8 ±0.0 |
| ✓ | × | 69.1 ±0.1 | 63.9 ±0.2 | 58.4 ±0.1 | 52.9 ±0.1 | 47.9 ±0.2 | 45.0 ±0.2 |
| ✓ | ✓ | **69.1** ±0.1 | **63.9** ±0.2 | **58.4** ±0.1 | **52.9** ±0.1 | **50.4** ±0.1 | **46.4** ±0.1 |

Test-N-Scene has improved. This is because the instruction style conversion model works on Scene-style instructions. In the third row, when only FST is added, compared with the first two rows, the performance of each column has improved, which indicates that our fast-slow thinking framework can improve the performance of all types of instructions, verifying its effectiveness. In the fourth row, when FST and ISC work together, the performance of Test-N-Scene reaches the best, verifying the collaborative effectiveness of FST and ISC.

**Experience library capacity $K$.** This experiment verifies the capacity saturation effect of experience library capacity $K$: whether insufficient storage limits generalization when $K$ is too small, and whether redundant experience causes surging computational overhead and low-quality interference when $K$ is too large, ultimately determining the optimal $K$ range. From Table 5, performance across all scenarios is lowest when $K = 20$, as the experience library fails to store key rules for OOD scenarios, restricting generalization. Test-R-Basic performs best at $K = 50$, since core experiences for basic instructions in residential scenes are sufficiently stored, and further increasing $K$ adds redundancy. Test-N-Basic and Test-N-Scene achieve optimal performance at $K = 100$, as OOD non-residential scenes require more generalized experiences. Performance declines at $K = 200$, possibly due to redundant experiences interfering with attention fusion. In summary, $K < 50$ leads to insufficient experience, while $K > 100$ causes redundancy. The optimal $K$ ranges from 50 to 100: we can use 100 for complex OOD scenarios and 50 for low-to-medium complexity scenarios.

## 3.3 CASE STUDY: BEFORE AND AFTER SLOW REASONING

To illustrate the qualitative advantages of the slow4fast architecture, we compare and analyze the situations of using fast reasoning alone and the interaction between slow and fast resoning. The instruction is: "*Leave the kitchen and take a right into the hallway. In the hall take the right into the den, then a left into the dining room. In the dining room stop next to the door*

Table 5: Analysis of the impact of $K$.

| $K$ | Test-R-Basic | | Test-N-Basic | | Test-N-Scene | |
|---|---|---|---|---|---|---|
| | SR↑ | SPL↑ | SR↑ | SPL↑ | SR↑ | SPL↑ |
| 20 | 61.3 ±0.2 | 54.7 ±0.1 | 47.5 ±0.2 | 44.0 ±0.0 | 43.6 ±0.2 | 40.2 ±0.2 |
| 50 | **65.4** ±0.1 | **64.9** ±0.1 | 54.9 ±0.0 | 47.4 ±0.1 | 47.5 ±0.1 | 44.0 ±0.3 |
| 100 | 62.0 ±0.2 | 57.6 ±0.0 | **60.2** ±0.2 | **52.0** ±0.1 | **48.5** ±0.1 | **45.6** ±0.1 |
| 200 | 63.1 ±0.1 | 64.0 ±0.0 | 58.0 ±0.1 | 51.6 ±0.0 | 46.3 ±0.3 | 44.7 ±0.0 |

*near the vent in the floor.*" The key spatial viewpoints are as follows: $V_1$: Inside the kitchen (starting point); $V_2$: Kitchen exit, connecting to the hallway; $V_3$: Hallway, with branches leading to $V_4$ (den entrance); $V_4$: Den entrance, leading to $V_5$ (left door to dining room); $V_6$: Bathroom, with a door leading to $V_7$ (dining room); $V_7$: Dining room, near the door and floor vent (destination).

**Initial Challenges.** The hallway has multiple branches, making it easy to go to the den and walk around in circles without prior experience; The visual feature "door near the vent" in the dining room is not prominent (the vent is small and partially obscured by a rug), making it easy to miss the target location during the first navigation attempt. At this stage, the experience library is empty. The fast-reasoning module relies solely on real-time vision and instructions, causing navigation errors.

**Navigation Trajectory (Initial Attempt).** $V_1$ (Inside Kitchen) → Leaves kitchen to $V_2$ (Kitchen Exit), turns right into the hallway ($V_3$) → Due to the dim lighting and multiple branches in the hallway, mistakenly selects the middle branch (not the correct right turn to the den), proceeds to the end of the hallway, and then turns back → Finds the correct right turn and enters the den ($V_4$) → After entering the den, fails to identify the left door to the dining room (partially blocked by a bookshelf) and wanders in a 1.2m circle → Finds the door and enters the bathroom ($V_6$) → Mistakenly identifies a common cabinet in the bathroom as the "door near the vent" and stops (fails to reach $V_7$). The total time consumed for navigation was 15 seconds, the navigation error reached 1.5 meters. The

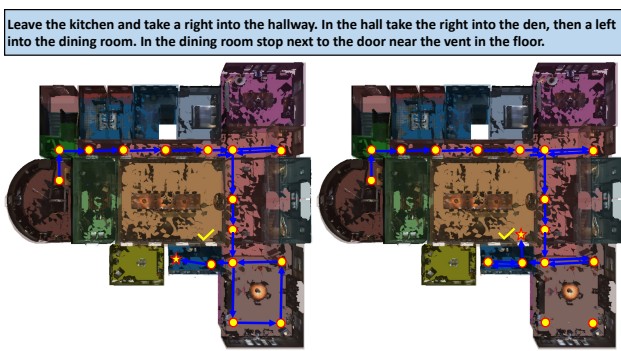

Leave the kitchen and take a right into the hallway. In the hall take the right into the den, then a left into the dining room. In the dining room stop next to the door near the vent in the floor.

Figure 3: **Case Study.** The left side shows the execution trajectory of the agent with **fast reasoning only**, while the right side displays the agent's execution trajectory **after slow reasoning optimization**. A check mark ($\checkmark$) indicates the destination (next to the door near the floor vent); A five-pointed star ($\star$) marks the final position reached by the agent.

core issue is the lack of prior knowledge of spatial features, such as the right turn to the den and the vent near the dining room door, leading to unnecessary detours and misidentifications.

**Experience Distillation (Post-Navigation Reflection).** The failure log from the first navigation is stored in the history repository, and the slow-reasoning module initiates a reflection process. The log is input into an LLM to generate experience $E_1$, which is stored in the experience library.

$S_t$ (*Scene Type*): residential-kitchen to dining room transition area.

$C_s$ (*Spatial Context*): Correct path in the hallway marked by a blue painting; the vent in the dining room is square and fully exposed, with a wooden door beside it.

$R_s$ (*Spatial Rules*): In the hallway, look for the blue painting on the second right branch; in the dining room, find the vent before the wooden door.

$T_n$ (*Navigation Strategy*): Upon reaching $V_3$ (Hallway), scan for the blue painting; in the dining room, locate the vent and the wooden door next to it.

After 4 iterations, the Experience Library has accumulated 6 similar experiences (e.g., "the effect of changing light in the hallway on branch identification," "reinforcing features of the left-turning door in the den"). During the 5th navigation, the fast-reasoning module is empowered by this experience.

**Navigation Trajectory (Post-Experience).** $V_1 \rightarrow V_2 \rightarrow V_3 \rightarrow$ Guided by the "blue painting" experience, quickly finds the second branch on the right (the correct path to the den) $\rightarrow$ Enters the den ($V_4$) and, based on the experience "the left door is next to a round desk," directly finds the left door to the dining room ($V_5$) $\rightarrow$ Enters the bathroom ($V_6$) and finds the right door leading to the dining room $\rightarrow$ Enters the dining room, locates the fully exposed square vent, and stops next to the wooden door beside it ($V_7$) (target). The total time taken was 8 seconds, representing a 46.7% reduction; the navigation error was minimized to 0.3 meters, an 80% reduction from the initial attempt; The improvements stem from the agent's enhanced experience, enabling it to quickly identify key features, leading to a more efficient path and precise target identification.

## 4 CONCLUSION

In this paper, we focus on the General Scene Adaptation for VLN (GSA-VLN) task, aiming to learn generalized navigation ability by introducing diverse environments and inconsistent intructions. Inspired by the fast-slow cognition systems, we propose the slow4fast-VLN, establishing a a dynamic interactive fast-slow reasoning framework. First, we receive input data, and the fast-reasoning module generating immediate navigation actions and storing memories. The slow-reasoning module analyzes these memories, extracts generalized experiences through deep reflection, and structurally stores them to optimize the fast module. This improves fast decision accuracy and generalization, enabling the system to adapt and execute navigation efficiently in unseen scenarios. Extensive experiments on GSA-R2R dataset demonstrate the superiorities of our method.

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

## A   APPENDIX

### A.1   RELATED WORK

**Vision-and-Language Navigation.** As a representative multi-modal embodied AI task, VLN requires an agent to combine human instructions and visual observations to navigate and locate targets in real-world scenes ( (Chen et al., 2022b; Anderson et al., 2021)). A challenge initially introduced through the R2R dataset ( (Anderson et al., 2018a)). Subsequently, various indoor VLN datasets have been developed ( (Krantz et al., 2020; Jain et al., 2019)), predominantly focusing on modifying textual inputs, including high-level object-oriented instructions ( (Yin et al., 2024)) and multimodal instructions ( (Hong et al., 2024)). Nevertheless, these datasets uniformly rely on identical scenes derived from the Matterport3D dataset ( (Chang et al., 2017)). Several research efforts have attempted to expand scene diversity by incorporating additional datasets ( (Chen et al., 2022a; Wang et al., 2023)). Yet, these methodologies exclusively serve as supplementary training resources, while evaluation remains confined to the original dataset splits with constrained environmental variety. In practice, agents must navigate previously unseen environments, posing a significant challenge for closed-set methods, which struggle to adapt to new settings with differing environmental and instructional contexts. Conversely, GSA-VLN ( (Hong et al., 2025)) introduces heterogeneous

environments and instruction types within the evaluation framework to thoroughly examine agent adaptability across both in-distribution (ID) and out-of-distribution (OOD) contexts.

**Adaptation Methods in VLN.** While single-scene adaptation remains an underexplored area in VLN research, several existing approaches provide relevant insights, including pre-exploration strategies (Wang et al., 2019). These methodologies can be broadly classified into two distinct categories. The first category encompasses optimization-driven approaches that dynamically adjust navigation model parameters within the target environment. Test-time adaptation techniques such as TENT (Wang et al., 2021) and SAR (Niu et al., 2023) refine model parameters through entropy minimization objectives, whereas Back-Translation (Wang et al., 2020) employs a trained speaker model to synthesize instructions for imitation learning purposes. The second category comprises memory-augmented approaches that explicitly maintain repositories of previously encountered locations and instructions to inform decision-making processes, exemplified by IVLN (Zhao et al., 2024) and RREx-BoT (Sigurdsson et al., 2023). For instance, TourHAMT (Krantz et al., 2023) integrates historical episode information as contextual embeddings, while OVER-NAV (Zhao et al., 2024) identifies instruction keywords within visual observations and maintains them in an Omnigraph structure for navigation assistance. Similarly, UniGoal (Yin et al., 2025) develops coherent 3D scene representations through graph structures, providing a framework for capturing and leveraging historical contextual information. More recently, LLM-powered VLN frameworks, including SE-VLN (Dong et al., 2025) and CogDDN (Huang et al., 2025), have exhibited remarkable zero-shot navigation capabilities, underscoring their promise for tackling scene adaptation challenges.

**Fast and Slow thinking.** The fast-slow thinking paradigm, rooted in dual-process theory from cognitive science (Kahneman, 2011), encompasses "fast intuition (System 1)" and "slow deliberation (System 2)" cognitive mechanisms. This framework provides a novel approach to address insufficient reasoning accuracy in AI models when tackling complex tasks, with extensive applications across visual-language model reasoning (Xiao et al., 2025), autonomous driving decisions (Xu et al., 2025), continual learning (Qi et al., 2024), large language model decision-making (Pan et al., 2024), visual agents (Sun et al., 2025), and robotic manipulation (Pan et al., 2024). Recent research indicates that through fast and slow cognition systems, humans can generate stable policies that enhance their adaptation to the open world. However, existing research employing this paradigm typically adopts rigid task categorization into simple versus difficult problems, implementing independent fast and slow systems respectively. This approach fails to establish the critical cognitive loop wherein slow thinking experience consolidates into fast thinking instincts. Slow thinking should transcend its role as a one-time problem-solving mechanism; its fundamental value resides in enabling accumulated experience to enhance the fast thinking system. The ultimate objective is developing intelligent agents capable of resolving the majority of problems through efficient fast thinking alone. Consequently, we propose a slow4fast-VLN framework designed to achieve the "slow reflection feeds fast action" closed-loop mechanism. Unlike traditional methods that treat fast-slow reasoning as independent mechanisms, our framework enables fast-slow interaction.

## A.2 TRAINING PIPLINE

---

**Algorithm 1** Interaction Training Process of Fast and Slow Reasoning

---

**Input:**
    Number of training iterations $T$
    Initial parameters of fast-reasoning model $\pi$ with $\theta$
    Initial Experience Library $\mathcal{M}_1 = \varnothing$
**Output:**
    Optimized model parameters $\theta_T$
    Final Experience Library $\mathcal{M}_T$
**Step 1: Initialization**
Initialize the fast-reasoning model parameters $\theta$
Initialize the reflective Experience Library $\mathcal{M}_1 = \varnothing$
**for** $t = 1$ to $T$ **do**
    **Step 2: Navigation Execution and Log Collection**
    Input environment observation $\mathcal{O}_t$ (initial viewpoint, visual features, and navigation instructions) into model $\pi$
    At each timestep $j$, model outputs action $A_j^s(t)$ and receives reward $r_j(t)$

---

Collect log data $\mathcal{L}(t_j)$ according to Equation (3)
Form the complete log set for this episode: $\mathcal{L}_{\text{ep}}(t) = \{\mathcal{L}(t_1), \ldots, \mathcal{L}(t_N)\}$
**Step 3: Context Construction and Experience Extraction**
Construct context data $\mathcal{X}(t)$ from log set $\mathcal{L}_{\text{ep}}(t)$
Generate structured experience $\mathcal{E}_{\text{new}}(t)$ using LLM reflection mechanism:
    $\mathcal{E}_{\text{new}}(t) = \mathcal{F}_{\text{LLM}}(\mathcal{P}(\mathcal{X}(t)))$
**if** LLM output parsing fails **then**
    Use default experience $\mathcal{E}_{\text{default}}$ instead of $\mathcal{E}_{\text{new}}(t)$
**end if**
**Step 4: Experience Library Update**
Update the Experience Library $\mathcal{M}_t$ based on $\mathcal{E}_{\text{new}}(t)$
**if** Experience $\mathcal{E}_i$ in $\mathcal{M}_t$ satisfies $\text{sim}(\mathcal{E}_{\text{new}}(t), \mathcal{E}_i) \geq \tau_{\text{update}}$ **then**
    Update the experience in the library: $\mathcal{M}_{t+1} = \text{Update}(\mathcal{M}_t, \mathcal{E}_{\text{new}}(t))$
**else**
    Add new experience to the library: $\mathcal{M}_{t+1} = \text{Add}(\mathcal{M}_t, \mathcal{E}_{\text{new}}(t))$
**end if**
**if** $|\mathcal{M}_{t+1}| > K$ **then**
    Delete the experience with the lowest quality from $\mathcal{M}_{t+1}$
**end if**
**Step 5: Experience Fusion and Model Optimization**
Retrieve relevant experiences $\mathcal{M}_{\text{sel}}(t+1)$ from $\mathcal{M}_{t+1}$ based on current context $\mathcal{X}_{\text{cur}}(t+1)$
Generate experience features $F_e(t+1)$ using experience encoder $\mathcal{G}_{\text{enc}}$
Generate enhanced visual features $F_{\text{fused}}(t+1)$ based on fusion strategy
Input $F_{\text{fused}}(t+1)$ into the agent to output navigation decisions $Y_{\text{enhanced}}(t+1)$
Optimize model parameters $\theta$ with the loss function $\mathcal{L}(\theta)$ of the navigation task:
    $\theta_{t+1} = \theta_t - \alpha \cdot \nabla_\theta \mathcal{L}(\theta_t, Y_{\text{enhanced}}(t+1), Y_{\text{gt}}(t+1))$
**end for**
**Return:**
    Optimized model parameters $\theta_T$
    Final Experience Library $\mathcal{M}_T$

## A.3 EXPERIENCE LIBRARY MANAGEMENT.

Let the Experience Library $\mathcal{M}$ be a finite set of experiences $\mathcal{M} = \{\mathcal{E}_1, \mathcal{E}_2, \ldots, \mathcal{E}_K\}$, where $K$ is the capacity limit of the Experience Library (i.e., the maximum number of experiences that can be stored). To avoid redundant experiences occupying storage resources and ensure the timeliness and effectiveness of experiences, this paper designs a similarity-based incremental update strategy with the following specific process:

**Similarity Calculation**: For a newly generated experience $\mathcal{E}_{\text{new}}$, first compute its comprehensive similarity $\text{sim}(\mathcal{E}_{\text{new}}, \mathcal{E}_i)$ with all existing experiences $\mathcal{E}_i \in \mathcal{M}$ in the Experience Library. Since experiences contain both discrete categorical features and continuous numerical features, a weighted fusion approach is used to calculate the comprehensive similarity:

$$\text{sim}(\mathcal{E}_{\text{new}}, \mathcal{E}_i) = \alpha \cdot \text{sim}_{\text{cat}}(\mathcal{E}_{\text{new}}, \mathcal{E}_i) + (1 - \alpha) \cdot \text{sim}_{\text{num}}(\mathcal{E}_{\text{new}}, \mathcal{E}_i), \tag{10}$$

where $\alpha \in [0, 1]$ is the weight coefficient (used to balance the contributions of the two types of features, set to 0.6 in experiments); $\text{sim}_{\text{cat}}$ is the similarity of discrete categorical features $(S_t, C_s, R_s, T_n)$, calculated using the Jaccard similarity coefficient; $\text{sim}_{\text{num}}$ is the similarity of continuous numerical features $(\gamma_d, \eta_s)$, calculated using cosine similarity.

**Experience Update Rules**: Let the similarity threshold for experience updates be $\tau_{\text{update}} \in [0, 1]$ (set to 0.7 in experiments). If there exists $\mathcal{E}_i \in \mathcal{M}$ satisfying $\text{sim}(\mathcal{E}_{\text{new}}, \mathcal{E}_i) \geq \tau_{\text{update}}$, then the two are judged to be similar experiences, and the existing experience $\mathcal{E}_i$ is incrementally updated (as shown in Equation 11); if no experience satisfying the threshold condition exists, then $\mathcal{E}_{\text{new}}$ is directly added to $\mathcal{M}$ as a new experience. If $|\mathcal{M}| > K$ after adding the new experience, then the experience with the lowest quality is deleted according to the experience quality assessment results (see Section A.3.1 for details) to maintain stable Experience Library capacity.

$$\mathcal{E}_i^{\text{new}} = \lambda \cdot \mathcal{E}_i + (1 - \lambda) \cdot \mathcal{E}_{\text{new}}, \tag{11}$$

where $\lambda \in [0, 1]$ is the update weight (set to 0.6 in experiments to balance the contributions of historical and new experiences), and simultaneously update the occurrence frequency $f_i$ of $\mathcal{E}_i$ to $f_i + 1$ to reflect the reuse value of this experience.

### A.3.1 EXPERIENCE QUALITY EVALUATION

To ensure the effectiveness of experiences in the Experience Library and avoid low-quality experiences affecting navigation decisions, this paper designs a multi-dimensional experience quality assessment function $\mathcal{Q}(\mathcal{E})$ to quantify the value of each experience $\mathcal{E}$. The evaluation metrics include four core dimensions: success rate, confidence, frequency, and timeliness, which are weighted and fused to obtain a quality score. The evaluation metric definitions are as follows:

Historical success rate $\eta_s$: The average success probability of navigation guided by this experience, directly taken from the $\eta_s$ dimension of experience $\mathcal{E}$ (see Equation (4));

Occurrence frequency $f_{\text{norm}}$: The normalized frequency of the experience in the Experience Library, $f_{\text{norm}} = \min(f/f_{\text{max}}, 1)$, where $f$ is the original frequency of the experience and $f_{\text{max}}$ is the maximum frequency of experiences in the Experience Library (set to 10);

Timeliness $\tau$: The freshness of the experience, $\tau = \exp(-\beta \cdot (t_{\text{now}} - t_{\text{last}}))$, where $t_{\text{now}}$ is the current time, $t_{\text{last}}$ is the last update time of the experience, and $\beta$ is the decay coefficient (controlling the decay rate of timeliness).

**Quality Score Calculation.** Fuse the four dimensions by weight to obtain the experience quality score $\mathcal{Q}(\mathcal{E})$:

$$\mathcal{Q}(\mathcal{E}) = w_1 \cdot \eta_s + w_2 \cdot f_{\text{norm}} + w_3 \cdot \tau, \tag{12}$$

where $w_1, w_2, w_3$ are the weight coefficients for each dimension, satisfying $w_1 + w_2 + w_3 = 1$. Through experimental validation, this paper sets the weights as $w_1 = 0.5$ (success rate as the core metric), $w_2 = 0.3$ (frequency reflects experience generality), and $w_3 = 0.2$ (timeliness avoids interference from outdated experiences).

**Low-Quality Experience Cleanup.** Periodically (every $T_{\text{clean}}$ iterations) sort the experiences in the Experience Library by quality, delete experiences with quality scores below the threshold $\tau_{\text{quality}}$, while retaining the top $K$ experiences by quality score (where $K$ is the Experience Library capacity), ensuring that the Experience Library always stores high-quality, highly relevant experiences:

$$\mathcal{M}_{\text{cleaned}} = \left\{ \mathcal{E} \in \mathcal{M} \mid \mathcal{Q}(\mathcal{E}) \geq \tau_{\text{quality}} \right\}_{1:K}, \tag{13}$$

where $\tau_{\text{quality}}$ is the quality threshold (set to 0.3 in experiments), and $\{\cdot\}_{1:K}$ represents taking the top $K$ experiences by quality score.

### A.4 QUANTITATIVE EXPERIMENTAL ON FAST-SLOW REASONING

To verify the core advantage of our interactive fast-slow reasoning architecture (referred to as the "serial empowerment method") in achieving a balance between efficiency and accuracy, this experiment uses the traditional threshold-switching fast-slow reasoning method (referred to as the "threshold-switching method") as the baseline, and conducts a comparative analysis based on quantitative metrics (efficiency, accuracy, robustness). The core logic of the threshold-switching method is as follows: a task difficulty threshold is preset, where fast reasoning (real-time vision + instruction matching) is only enabled for simple tasks, and slow reasoning (global scene reasoning + path planning) is triggered for complex tasks. In contrast, the proposed serial empowerment method constructs an experience library through the slow reasoning module, and during the real-time phase, navigation is executed solely by fast reasoning, dynamically invoking experiences, without switching to slow reasoning throughout the process.

Table 6: Quantitative experimental results on fast-slow reasoning.

| Task Difficulty | Method | Avg. Time (s) | Avg. FLOPs (G) | Success Rate (%) | Avg. Nav Error (m) |
|---|---|---|---|---|---|
| Easy | Baseline | 8.2±0.5 | 12.5±0.8 | 84.0±1.2 | 0.21±0.03 |
| | Ours | 7.9±0.4 | 10.2±0.6 | 88.5±1.0 | 0.18±0.02 |
| Medium | Baseline | 12.5±0.8 | 28.6±1.5 | 75.0±2.3 | 0.45±0.05 |
| | Ours | 9.8±0.6 | 11.5±0.7 | 82.0±1.8 | 0.32±0.04 |
| Hard | Baseline | 20.3±1.2 | 56.8±2.1 | 52.0±3.5 | 0.68±0.08 |
| | Ours | 12.1±0.9 | 13.2±0.9 | 58.0±2.6 | 0.61±0.06 |

**Experimental Scenarios and Task Library Construction.** To cover typical application scenarios and task difficulties of indoor navigation, the experiment selects three standardized indoor environments (residential homes, office spaces, hotel rooms) from the GSA-R2R (Hong et al., 2025) dataset. Each environment includes 20 differentiated navigation tasks, totaling 60 tasks. Based on the three dimensions of "obstacle level," "target interference," and "path complexity," the tasks are categorized into three difficulty levels, defined as follows:

*Simple tasks* (20): No obstructions, straightforward paths, and unique target features, e.g., "Living room to dining table in a residential home."

*Moderate tasks* (20): One temporary obstruction (e.g., chair, suitcase), with no similar target interference, e.g., "Office workspace to break room, with a temporarily placed file box at the door."

*Complex tasks* (20): Two or more fixed obstructions (e.g., bookshelf, wardrobe) with similar target interference (e.g., "white refrigerator" vs. "white storage cabinet"), e.g., "Hotel room from desk to minibar, distinguish between minibar and similar-looking decorative cabinet."

**Baseline Method (Threshold Switching Approach).** Referring to classic fast and slow reasoning frameworks such as RFST (Zhu et al., 2024) and FAST (Sun et al., 2025), the core implementation logic is as follows:

*Difficulty Threshold Setting:* The task is considered a complex task and triggers a switch to slow reasoning when the combined thresholds of navigation error > 0.8m and decision stagnation time > 2s are exceeded.

*Fast Reasoning Module:* Based on the DUET Chen et al. (2022b) architecture, it inputs real-time visual features (extracted by ViT-B/16) and instruction text features, outputting instantaneous navigation actions (e.g., move forward, turn, stop), without an experience reuse mechanism.

*Slow Reasoning Module:* Utilizes the Llama3.2-vision large language model, which receives the full-scene visual feature sequence and instruction, performs multi-step reasoning to generate global path planning, and then breaks it down into local navigation actions.

**Our Method (Serial Empowerment Approach).** The agent uses the DUET architecture for the fast reasoning module, consistent with the Baseline, to ensure a fair comparison. For the slow reasoning module, the agent also employs Llama3.2-vision, which processes the historical memories generated by fast reasoning, generates experiences, and outputs optimized navigation actions. The core difference lies in the fact that no switch to slow reasoning is needed throughout the process. Instead, decision optimization is achieved through lightweight experience retrieval and feature fusion, avoiding the high computational cost and reasoning delays typically associated with slow reasoning.

We selected four core metrics: Average Navigation Time (Avg. Time): the arithmetic mean of the completion times for all tasks, primarily reflecting the impact of slow reasoning switching delay on complex tasks; Average Computational Load (Avg. FLOPs): the average total floating-point operations during navigation, including feature extraction and decision-making inference, primarily measuring hardware computational cost; Task Success Rate (Success Rate); and Average Navigation Error (Avg. Nav Error). The results are shown in Table 6.

The results show that in terms of efficiency, as the task difficulty increases, the difference in Avg. Time and Avg. FLOPs between Ours and Baseline gradually becomes larger. In complex tasks,

Ours has an Avg. Time that is approximately 40% lower and Avg. FLOPs that are about 77% lower than Baseline. The core reason is that Baseline frequently switches to slow reasoning, while Ours relies only on lightweight experience retrieval. In terms of accuracy advantages: both methods have similar accuracy, indicating that Ours can improve accuracy in complex tasks without sacrificing efficiency performance. In terms of robustness, Ours's Avg. Nav Error is lower than Baseline across tasks of all difficulty levels, which demonstrates that our method has stronger generalization ability and is more suitable for the generalization required by GSA-VLN tasks.

## A.5 VISUALIZATION

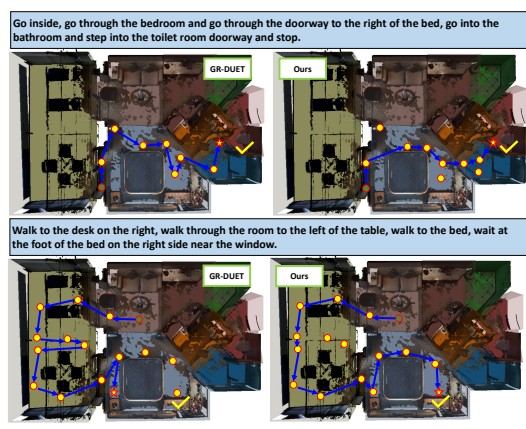

Figure 4: Predicted trajectories of GR-DUET (left) and our method (right). A check mark (✓) indicates the destination; A five-pointed star (⋆) marks the final position reached by the agent.

We visualized and compared the predicted trajectories of GR-DUET (Hong et al., 2025) and our method (see Fig. 4): In the scenarios shown in the first row, both methods reached the correct destination, but GR-DUET exhibited inefficient, excessively long trajectories with unnecessary detours due to its lack of accumulated scene experience. In the second row, GR-DUET incorrectly stopped to the left of the bed-mistaking a bedside lamp for a window and failed to reach the target, with its trajectory also being overly redundant. By contrast, our method efficiently arrived at the window to the right of the bed. This advantage stems from the scene spatial rules (e.g., "window on the right wall of the bed") and the dynamic experience loop in our slow-thinking module. It helps avoid detours by using the experience library and accurately identifies targets through a combination of real-time visual observations and scene priors. The gap arises because our "execution-reflection-empowerment"

mechanism allows the agent to learn from practice, while GR-DUET, relying on static rules without experience integration, misjudgments in complex situations.

## A.6 PERFORMANCE OF LLM-BASED METHODS

We incorporated several other LLM-based methods to explore whether they can adapt to different instruction styles without additional adaptation processes, as shown in the table 7. Unlike them, among these three, the LLM is the direct decision-making entity, and its core task is to output actions or reasoning for the current step. Our focus, however, is that the LLM does not directly make navigation decisions but empowers the decision-making module through knowledge distillation, while emphasizing the dynamic accumulation and iterative optimization of experience, making it more suitable for the GSA-VLN task. The results show that compared with our method, the other three methods perform poorly on the GSA-R2R task. Although large language models can handle different styles of instructions, they face difficulties in environmental adaptation required by GSA-VLN, especially in processing visual information and interacting with persistent environments.

Table 7: Comparison of LLM-based methods on GSA-R2R.

| Methods | Test-N-Basic | | Test-N-Scene | | Test-N-User | |
|---|---|---|---|---|---|---|
| | SR↑ | SPL↑ | SR↑ | SPL↑ | SR↑ | SPL↑ |
| MapGPT(Chen et al., 2024) | 34.17 | 29.72 | 24.67 | 22.62 | 23.17 | 20.80 |
| NavCoT(Lin et al., 2025) | 36.67 | 34.46 | 29.00 | 25.93 | 26.33 | 24.47 |
| NavGPT-2(Zhou et al., 2024) | 63.50 | 47.26 | 56.67 | 43.34 | 47.00 | 36.86 |
| **Ours** | **75.36** | **70.97** | **55.10** | **48.22** | **58.65** | **53.40** |

**a. Role Guidance Text:**

You are an expert in the navigation field, specializing in analyzing the navigation behaviors of agents and extracting reusable abstract experiences. Your core task is to deeply understand the spatial relationships, visual context, and behavioral patterns throughout the navigation process, and derive general rules and strategies from them that can guide future navigation decisions.

**Task Definition:**
Please conduct an in-depth analysis from the following 5 core dimensions:
**1. Scene Recognition Analysis**
- Spatial Type Inference: Analyze what type of spatial environment you are currently in? (Living room, corridor, kitchen, etc.)
- Spatial Feature Extraction: Identify key features of the space, such as door positions, passage directions, and functional area divisions
- Spatial Layout Understanding: Comprehend the overall layout and functional zoning of the space
**2. Spatial Relationship Analysis**
- Topological Relationships: Analyze the place relationships and spatial connection patterns between adjacent nodes
- Distance and Direction: Understand how relative azimuth angles and distances reflect the spatial layout
- Spatial Transition Patterns: Identify the laws of transition from one spatial type to another
**3. Visual Context Analysis**
- Visual Landmark Recognition: Identify key visual landmarks (doors, windows, furniture, electrical appliances, etc.)
- Visual-Spatial Association: Analyze how visual information guides spatial navigation decisions
- View Orientation Analysis: Understand the relationship between view orientation and target direction
**4. Behavioral Analysis**
- Decision-Making Pattern: Analyze the agent's decision-making logic and strategies
- Action Sequence: Identify the laws and patterns of the action sequence
- Confidence Analysis: Evaluate the confidence and reliability of decisions
**5. Experience Extraction**
- Spatial Rules: Extract general spatial navigation rules
- Navigation Strategies: Summarize effective navigation strategies and methods
- Improvement Suggestions: Propose directions for improving future navigation decisions

**b. Output Format:**
Please return the analysis results in **JSON format**, ensuring that the fields correspond to the experience dimensions one by one:
{"scene_type": "Description of scene type",
"spatial_context": "Description of spatial context",
"spatial_rule": "Extracted spatial rules",
"navigation_strategy": "Description of navigation strategy",
"place_relationship": "Description of place relationship",
"visual_context": "Description of visual context",
"visual_landmarks": "Key visual landmarks",
"visual_orientation": "Description of view orientation",
"improvement_suggestion": "Improvement suggestions"}

**c. Navigation Context：**

**Instruction:** {instruction}; **Episode ID:** {episode_id}; **Success Status:** {success_status};
**Overall Metrics:** {spl_value, navigation_error_value, action_steps_value, trajectory_length_value, ndtw_value, sdtw_value, cls_value}
**Detailed Step-by-Step Analysis:** {
Step 1 ($t_1$={time_1}, j_seq={seq_1}):
Current Position: {current_position_1}
Local Topology: Reachable Neighbors: {reachable_neighbors_1, reachable_neighbors_2, reachable_neighbors_3},
Relative Azimuth Angle: {relative_azimuth_angle_1, relative_azimuth_angle_2, relative_azimuth_angle_3},
Straight-Line Distance: {straight_line_distance_1, straight_line_distance_2, straight_line_distance_3},
Selected Action: {selected_action_1},
Visual Description: {visual_description_1},
Stop Probability: p_stop = {stop_probability_1},
….
Step N

Figure 5: Prompt Template for the Reflection Module.

**a.  Role Guidance Text:**

You are a professional instruction style conversion expert. Please convert the following personalized user-style instruction into a concise basic-style instruction.

**Conversion rules:**
1. Remove personalized expressions:
  - Delete: hey, could you, please, thanks, awesome, cool
  - Delete: darling, sweetheart, dear, lovely, wonderful
  - Delete: like, totally, amazing, incredible, fantastic
  - Delete: fascinating, intriguing, precisely, logically
  - Delete: that would be, when convenient, for me

2. Vocabulary standardization:
  - go to → go
  - walk to → walk
  - turn around → turn
  - stop at → stop
  - head to → go
  - move to → walk
  - proceed to → go
  - navigate to → walk
  - execute a turn → turn
  - halt at → stop

3. Syntax simplification:
  - Convert interrogative sentences into declarative sentences
  - Remove polite expressions and emotional words
  - Retain core action and direction information

4. Retain navigation information:
  - Keep all action instructions (go, turn, walk, stop, etc.)
  - Keep all direction information (left, right, forward, backward, etc.)
  - Keep all target locations (kitchen, bathroom, desk, etc.)

5. Output requirements:
  - Use standard navigation vocabulary
  - Sentences are concise and direct
  - No emotional overtones
  - Directly express navigation intent

**b. Conversion examples:**
- "Hey, could you go to the kitchen please?" → "Go to the kitchen"
- "Darling, walk to the bathroom for me" → "Go to the bathroom"
- "Like, turn left, that would be amazing!" → "Turn left"
- "Kindly proceed to the workstation" → "Go to the desk"
- "Fascinating, execute a rightward rotation" → "Turn right"

**c.  Input User-style instruction:** {user_instruction}

**d.  Output：** {Basic-style instruction}

Figure 6: Prompt Template for Converting User-Style to Basic-Style.

**a. Role Guidance Text:**

You are a professional instruction style conversion expert. Please convert the following personalized scene-style instruction into a concise basic-style instruction.

**Conversion rules:**

1. Processing of modifiers :
   - Delete: precisely, carefully, cautiously, steadily
   - Delete: utmost, greatest, maximum, optimal
   - Delete: steadfast, unwavering, consistent
   - Delete: with care, with precision, with caution
   - Delete: while maintaining, whilst ensuring

2. Vocabulary standardization:
   - proceed/advance → go
   - navigate/traverse → walk
   - execute/perform → turn
   - halt/cease → stop
   - circumvent/avoid → go around
   - maintain/keep → keep
   - approach/advance to → go to
   - depart/leave → leave
   - position/place → place
   - orient/face → turn to face

3. Syntax simplification:
   - Convert interrogative sentences into declarative sentences
   - Remove polite expressions and emotional words
   - Retain core action and direction information

4. Retain navigation information:
   - Keep all action instructions (go, turn, walk, stop, etc.)
   - Keep all direction information (left, right, forward, backward, etc.)
   - Keep all target locations (kitchen, bathroom, desk, etc.)

5. Output requirements:
   - Use standard navigation vocabulary
   - Sentences are concise and direct
   - No emotional overtones
   - Directly express navigation intent

**b. Conversion examples:**

- "Proceed forward, circumventing the workstation with the utmost care" → "Go around the desk"
- "Execute a leftward rotation while maintaining a steadfast course" → "Turn left"
- "Navigate through the corridor with precision" → "Walk through the hallway"
- "Halt precisely at the workstation and await further instructions" → "Stop at the desk"
- "Maintain forward progression while avoiding the obstacle" → "Go forward around the obstacle"
- "Advance to the designated location with caution" → "Go to the location"
- "Traverse the facility while maintaining optimal positioning" → "Walk through the building"
- "Position yourself at the entrance with precision" → "Go to the door"
- "Execute a clockwise rotation to face the target" → "Turn right to face the target"
- "Retreat to the previous position while maintaining orientation" → "Go back to the previous place"

**c. Input Scene-style instruction:** {scene_instruction}

**d. Output：** {Basic-style instruction}

Figure 7: Prompt Template for Converting Scene-Style to Basic-Style

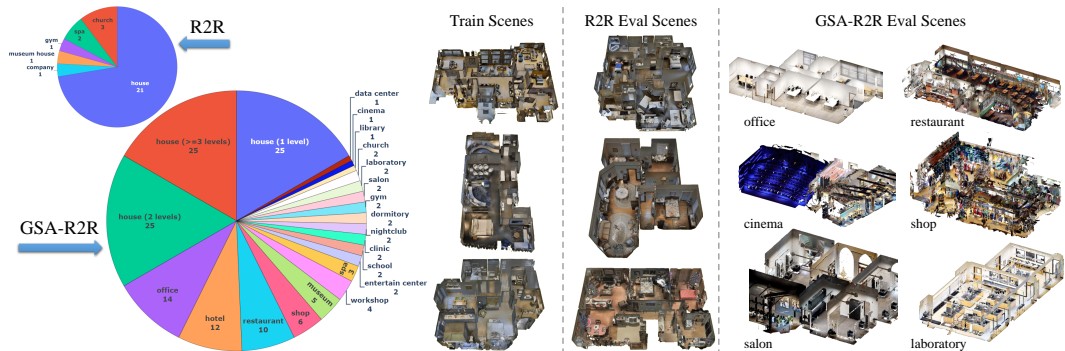

Figure 8: **Left:** Building type counts in R2R and GSA-R2R. **Right:** Comparison of buildings in R2R and GSA-R2R. Unlike R2R, where evaluation scenes are similar to the training set, GSA-R2R includes a more diverse mix of both in-distribution (ID) and out-of-distribution (OOD) data. This figure is from GR-DUET (Hong et al., 2025).

Table 8: Compared to the evaluation part of existing datasets in embodied navigation tasks. †: TouchDown is an outdoor dataset in New York City, which makes it hard to define scene types. This table is from GR-DUET (Hong et al., 2025).

| Dataset | Source | Scenes | | Path | Instructions | | Vocab Size | |
|---|---|---|---|---|---|---|---|---|
| | | Num | Type | Num | Num | Type | All | Unseen |
| R2R (Anderson et al., 2018b) | MP3D | 29 | 6 | 2,174 | 6,522 | 1 | 1,946 | 545 |
| R4R (Jain et al., 2019) | MP3D | 29 | 6 | 5,026 | 45,234 | 1 | 1,230 | 221 |
| RxR-en (Ku et al., 2020) | MP3D | 29 | 6 | 4,201 | 8,636 | 1 | 4,789 | 1,387 |
| CVDN (Thomason et al., 2020) | MP3D | 29 | 6 | 2,741 | 2,741 | 1 | 1,559 | 490 |
| TouchDown† (Chen et al., 2019) | Google Street View | 1 | - | 2,800 | 2,800 | 1 | 3,104 | 759 |
| RobustNav (Chattopadhyay et al., 2021) | ROBOTHOR | 15 | 1 | 1,800 | - | - | - | - |
| PASTURE (Gadre et al., 2023) | ROBOTHOR | 15 | 1 | 2,520 | 2,520 | 1 | 123 | 111 |
| GSA-R2R (Hong et al., 2025) | MP3D & HM3D | 150 | 20 | 90,000 | 90,000 | 7 | 4,337 | 2,905 |

## A.7 MORE DETAILS OF DATASET

The limited number and diversity of existing VLN datasets make them unsuitable for the GSA-VLN task. For instance, the most widely used VLN dataset, R2R (Anderson et al., 2018a), includes 90 building-scale scenes from the Matterport3D (MP3D) dataset (Chang et al., 2017) , with only 29 scenes for evaluation and most of them are residential houses. Each building contains a limited number of paths, ranging from a maximum of 100 to as few as 6, with each path paired with three natural language instructions that share a similar concise and plain style. GSA-R2R dataset not only provides sufficient data to allow continuous agent optimization to test their specialization, but also includes a diverse range of building types and instruction types to evaluate the agent generalization to various application scenarios. As shown in Table 8 and Figure 8.

## A.8 LIMITATION

Knowledge derived from slow reasoning is implicitly encoded into the weights of the policy network through retraining. This "black-box" approach makes the learned experiences difficult to interpret. A promising research direction is to enable the slow reasoner to generate an explicit, structured knowledge base (such as a semantic map or a knowledge graph). In this way, the fast reasoner can directly query this knowledge base during navigation to make more rational decisions.

## A.9 LLM DISCLAIMER

In this paper, we use LLM for polishing and reading the article, aiming to improve efficiency and readability. Specifically, in the method section of the article, we use LLM for logical sorting and checking the normativity of formulas.

