# OpenReview forum: "Towards open environments and instructions: general vision-language navigation via fast-slow interactive reasoning"
_ICLR.cc/2026/Conference — ICLR 2026 Conference Withdrawn Submission_

### Official Review · Reviewer_BNmG · 2025-10-15

**Soundness:** 3
**Presentation:** 3
**Contribution:** 3
**Rating:** 6
**Confidence:** 3

**Summary:**

This work proposes a fast–slow navigation framework. The fast module generates actions based on real-time observations and instructions, while the slow reasoning module refines the historical data into structured knowledge by analyzing key successes and failures. The framework appears both practical and intuitively sound.

**Strengths:**

Current navigation methods often lack robustness and perform poorly in out-of-distribution (OOD) environments. The dual-system framework may offer a promising solution to this challenge by enhancing the generalization ability of navigation systems.

**Weaknesses:**

I am looking for some experiments testing on real mobile robots. the current testing results is trying to show it has good performance on out of distribution data. I think this is the key motivation of the paper. how about training with offline dataset but testing in real world. like is it solving the simtoreal problem simultaneously?

**Questions:**

1. In the section “Experience Library Capacity (K)”, I wonder whether the choice of K depends on the type of scene. For instance, dynamic environments such as streets change rapidly, while more static settings like offices remain stable. The update frequency and the length of K of the slow system might therefore need to be adapted to the scene dynamics.
2. Have you tested the framework on real mobile robots to validate its effectiveness beyond simulation?
3. Have you observed any hallucinations in the system—either visual hallucinations in perception or LLM-related hallucinations in text-based reasoning or instruction interpretation?

---

### Official Review · Reviewer_ejei · 2025-10-30

**Soundness:** 3
**Presentation:** 2
**Contribution:** 2
**Rating:** 4
**Confidence:** 4

**Summary:**

This paper tackles the General Scene Adaptation for Vision-Language Navigation (GSA-VLN) task, aiming to improve generalization to unseen environments and diverse instruction styles. The authors propose Slow4Fast-VLN, an interactive fast–slow reasoning framework inspired by dual-process cognition. A DUET-based policy handles fast reasoning, while a LLaMA3.2-vision module conducts slow reflection to extract structured experience knowledge. Experiments on GSA-R2R show improvements over GR-DUET.

**Strengths:**

- The paper focuses on open-world generalization in VLN, a direction gaining traction in embodied AI and multimodal reasoning. The use of dual-process cognition (fast and slow reasoning) to structure VLN decision processes is interesting and aligns with ongoing efforts to bring cognitive inspiration into LLM-augmented agents.

- The use of GSA-R2R, which extends R2R with out-of-distribution scenes and diverse instruction styles, is suitable for testing generalization.

- The paper includes ablations (FSR and ISC modules, experience library capacity) and a qualitative case study that helps readers understand the intuition behind the slow-reflection process.

**Weaknesses:**

- While the fast–slow reasoning framework is well-motivated conceptually, its technical realization mainly combines existing components (DUET backbone + LLM-based reflection + attention fusion). The interaction between fast and slow reasoning is implemented as an experience retrieval and attention fusion module, which feels incremental rather than a fundamentally new reasoning paradigm.

- The reported improvements over GR-DUET are small (typically +1–2% SR), which may fall within noise levels given the scale of evaluation. There is no clear demonstration that these gains stem from the proposed fast–slow interaction, rather than from using a larger or more capable LLM (LLaMA3.2-vision).

- The claim of interactive reasoning remains somewhat concerns: the slow reasoning appears to be executed offline between episodes, not in real time. It is unclear whether the learned experience embeddings truly generalize to unseen scenes or simply act as additional memory augmentation.

- The instruction style conversion module seems to rely heavily on LLM rewriting; however, the paper lacks analysis on how conversion quality affects navigation accuracy. The mechanism might introduce confounding variables (e.g., LLM paraphrasing fidelity) that are not controlled or discussed.

**Questions:**

- Can the authors show quantitative evidence that the slow reasoning directly improves future fast decisions (e.g., ablations without memory fusion)?

- How is the LLM’s contribution isolated from the architecture itself? Would smaller models achieve similar gains?

- Why are more recent LLM-based VLN methods not included as baselines?

---

### Official Review · Reviewer_1e2w · 2025-11-01

**Soundness:** 2
**Presentation:** 3
**Contribution:** 2
**Rating:** 2
**Confidence:** 5

**Summary:**

This paper introduces **Slow4Fast-VLN**, a novel *fast–slow interactive reasoning framework* for the **General Scene Adaptation Vision-Language Navigation (GSA-VLN)** task. Traditional VLN methods assume a closed world where training and testing share similar environments and instruction styles, which limits adaptability to unseen domains. The authors design an interactive dual-process architecture inspired by human cognition: a **fast reasoning module** (a policy network executing actions in real time) and a **slow reasoning module** (an LLM-based reflection system that analyzes history logs, extracts generalizable experiences, and refines fast policies). Additionally, the paper proposes an **instruction style conversion mechanism** using Chain-of-Thought prompting to normalize diverse user and scene instructions into a unified format. Experiments on the **GSA-R2R** benchmark demonstrate performance gains across in-distribution (residential) and out-of-distribution (non-residential) settings, showing improvements over GR-DUET and other adaptation baselines. Ablation, qualitative case studies, and sensitivity analyses further illustrate the benefits of the fast–slow interaction and experience library design.

**Strengths:**

**Innovative Cognitive Framework:**
The idea of *interactive* fast–slow reasoning—where System 2 (slow reflection) directly updates and empowers System 1 (fast policy)—is an elegant adaptation of dual-process cognition to embodied VLN. Unlike previous “parallel” designs, this model establishes a genuine feedback loop, forming a “slow-reflection-feeds-fast-action” mechanism.

**Integration of LLM Reflection in Embodied Learning:**
The slow reasoning module uses structured Chain-of-Thought (CoT) prompts to distill experiences from navigation logs (Eq. 6). This implementation is well-motivated, and the structured experience vector E = [St, Cs, Rs, Tn, ηs, f] ⊺ provides an interpretable schema for extracting transferable knowledge from trajectory histories.

**Comprehensive Evaluation:**
Results across basic, scene-style, and user-style instructions (Tables 1–3) are extensive. The model improves SR by +1.5 % (ID) and +2.2 % (OOD) over GR-DUET under basic instructions, and consistently outperforms all baselines across five user personas and scene-style inputs. Ablations on both the **Fast–Slow Reasoning (FSR)** and **Instruction Style Conversion (ISC)** components are systematic (Table 4).

**Clarity of Case Study:**
The qualitative example (Figure 3) clearly demonstrates how slow reasoning refines spatial understanding (e.g., identifying the “blue painting” landmark) and converts failure logs into structured spatial strategies that subsequently shorten trajectory length and reduce error by 80 %.

**Relevance to Open-World VLN:**
By addressing both *scene generalization* and *instruction variability*, the paper aligns well with recent goals in embodied AI for open-environment robustness, bridging cognitive modeling and LLM-based navigation.

**Weaknesses:**

**1. Conceptual Novelty—Limited Algorithmic Depth:**
While the interactive fast–slow loop is well presented, it remains conceptually similar to existing *dual-process* or *meta-reflection* frameworks in VLN (e.g., MiC 2024, VLN-Copilot 2024, CogDDN 2025; SE-VLN 2025; NavCoT 2025) and in general LLM reasoning (e.g., Fast-Slow LLM 2025). The “interaction” here mainly consists of attention-based fusion (Eq. 7–9) between retrieved experience embeddings and visual features—a modest engineering extension rather than a fundamental algorithmic innovation.

**2. Motivation and Design Ambiguity:**
The paper justifies the fast–slow split primarily through analogy to human cognition but provides limited empirical or theoretical grounding for *why* this two-stage interaction specifically enhances adaptation. It remains unclear how the “slow reasoning” differs functionally from a conventional memory-augmented replay or knowledge distillation step. The “instruction style conversion” component also appears ad hoc, with little ablation on prompt reliability or conversion accuracy.

**3. Missing Discussion of Fast–Slow Discrepancy:**
The fast module reasons over latent embeddings, whereas the slow module reasons over symbolic textual reflections derived from visual descriptions. The discrepancy between these representation spaces (latent vs. linguistic) is never analyzed—raising questions about alignment, error propagation, and information loss during experience encoding.

**4. Incomplete Theoretical and Computational Analysis:**
Key parameters (e.g., retrieval threshold τ or attention dimension d) are mentioned without sensitivity analysis. The computational overhead of LLM reflection (frequency, token cost) is not reported, leaving unclear whether the method is practical for real-time or large-scale deployment.

**5. Writing and Presentation Issues:**
The paper occasionally repeats background concepts (System 1/2 definitions, cognitive analogy) and lacks concise mathematical exposition. Equations (2–9) are descriptive rather than formally derived, and variable names are sometimes inconsistent. The introduction would benefit from clearer positioning relative to prior fast–slow reasoning work in embodied contexts.

**Questions:**

1. **Interaction Frequency:** How often is slow reasoning invoked during training? After each episode or periodically? How do you control LLM inference cost versus learning benefit?
2. **Representation Alignment:** How is semantic consistency ensured between textual experiences (E) and visual feature embeddings (Fv)?
3. **Instruction Style Conversion Robustness:** How often does the system reject LLM-converted instructions due to low confidence?
4. **Comparison to Mic, VLN-copilot, SE-VLN, and CogDDN:** These works also use cognitive dual systems. How does Slow4Fast differ beyond experience-attention fusion?

---

### Note · Authors · 2025-11-12

I have read and agree with the venue's withdrawal policy on behalf of myself and my co-authors.